# The Agronomic Traits, Alkaloids Analysis, FT-IR and 2DCOS-IR Spectroscopy Identification of the Low-Nicotine-Content Nontransgenic Tobacco Edited by CRISPR–Cas9

**DOI:** 10.3390/molecules27123817

**Published:** 2022-06-14

**Authors:** Jianduo Zhang, Qun Zhou, Dongheyu Zhang, Guangyu Yang, Chengming Zhang, Yuping Wu, Yong Xu, Jianhua Chen, Weisong Kong, Guanghui Kong, Jin Wang

**Affiliations:** 1Yunnan Key Laboratory of Tobacco Chemistry, Research and Development Center, China Tobacco Yunnan Industrial Co., Ltd., Kunming 650231, China; zjdbio@163.com (J.Z.); yanggy@ynzy-tobacco.com (G.Y.); zhangcm@ynzy-tobacco.com (C.Z.); xuyong@ynzy-tobacco.com (Y.X.); chenjianhuabj@163.com (J.C.); kongws@ynzy-tobacco.com (W.K.); 2Department of Chemistry, Tsinghua University, Beijing 100084, China; zhouqun@mail.tsinghua.edu.cn; 3Department of Electrical Engineering, Tsinghua University, Beijing 100084, China; z_d_h_y@163.com; 4Yunnan Academy of Tobacco Agricultural Science, Kunming 650031, China; ypwumm@163.com (Y.W.); 13908776036@163.com (G.K.)

**Keywords:** gene-edited tobacco, BBLs, FT-IR, 2DCOS-IR, identification

## Abstract

In this study, the agricultural traits, alkaloids content and Fourier transform infrared spectroscopy (FT-IR) and two-dimensional correlation infrared spectroscopy (2DCOS-IR) analysis of the tobacco after Berberine Bridge Enzyme-Like Proteins (BBLs) knockout were investigated. The knockout of BBLs has limited effect on tobacco agricultural traits. After the BBLs knockout, nicotine and most alkaloids are significantly reduced, but the content of myosmine and its derivatives increases dramatically. In order to identify the gene editing of tobacco, principal component analysis (PCA) was performed on the FT-IR and 2DCOS-IR spectroscopy data. The results showed that FT-IR can distinguish between tobacco roots and leaves but cannot classify the gene mutation tobacco from the wild one. 2DCOS-IR can enhance the characteristics of the samples due to the increased apparent resolution of the spectra. Using the autopeaks in the synchronous map for PCA analysis, we successfully identified the mutants with an accuracy of over 90%.

## 1. Introduction

Nicotine is the primary alkaloid of tobacco, and its main biological function is defense against herbivores [1,2,3,4,5,6]. In the tobacco industry, nicotine is an important indicator of commercial quality of tobacco and cigarettes, and it is also the primary addictive and harmful substance in cigarettes [7,8]. Therefore, nicotine reduction was considered as one feasibility strategy for tobacco control. Nicotine is biosynthesized in the roots of tobacco, transferred, and accumulated in the vacuoles of tobacco leaves. It is composed of a pyridine ring and a pyrrole ring, and synthesized by independent metabolic pathways [1,6]. The last biosynthesis reaction of nicotine, the couple of the pyridine ring and the pyrrole ring, is believed to be catalyzed by Berberine Bridge Enzyme-Like Proteins (BBLs) [1,4,6]. The BBL gene family in tobacco has been identified as six homologs, BBLa, BBLb, BBLc, BBLd1, BBLd2, and BBLe. BBLs suppression by RNAi and knockout with ethyl methanesulfonate (EMS) resulted in a significant reduction of nicotine in tobacco [7,9]. In recent work, when six BBLs genes were knocked out via a CRISPR–Cas9-based knockout strategy, the obtained tobacco was nicotine-free [7]. There are some reports on the regulation of nicotine content by the BBL gene. However, the tobacco agronomic traits and the alkaloid content changes after BBLs knockout are rarely systematically studied. On the other hand, in gene editing research, in addition to molecular biology screening methods, it is also very necessary to construct a rapid and high-throughput screening and identification method based on chemical analysis.

Fourier transform infrared spectroscopy (FT-IR), providing an application of the most available, simple, and rapid method, can be an efficient alternative for measuring metabolites [10,11,12]. The FT-IR method produces a spectrum that provides complex data which can be used to thoroughly describe the chemical characteristics of a sample. Although the IR spectrum pattern complexity is visually difficult to interpret, the chemometrics method can be used to resolve this problem [10]. Moreover, the appearance of a two-dimensional correlation (2DCOS) infrared spectroscopy [13,14] can significantly improve the resolution of the infrared spectra based on the perturbation applied to the samples. Its ability to analyze complex samples, such as traditional Chinese medicines, plants, and foods, has been proven [15,16,17,18,19]. Hence, the high-resolution 2DCOS-IR may be used for the screening and identification of gene-editing materials. 

In our previous research, the low-nicotine-content nontransgenic tobacco was obtained by the knockout of the key nicotine synthesis gene, BBL, via gene editing with CRISPR–Cas9. In this paper, the agricultural traits, alkaloids content, FT-IR, and 2DCOS-IR spectroscopy analysis of the tobacco after BBLs knockout were investigated for the first time. Additionally, chemometric analyses of the FT-IR and 2DCOS-IR data were performed to identify the BBLs knockout tobacco.

## 2. Materials and Methods

### 2.1. Plant Materials

The tobacco seeds were derived from the tobacco variety Hongda flue-cured tobacco and its counterpart homozygous mutant, which is a knockout of BBLa, BBLb, BBLc, BBLd1, BBLd2, and BBLe by CRISPR–Cas9, as described by Schachtsiek et al. [7]. All tobacco plants were grown in an open field in Kunming, China. Tobacco samples were collected at the flower-bud appearing stage. In order to eliminate the influence of individual differences, the sampling used the mixed sample method, and the leaf or root samples of three tobacco plants, were mixed into one sample. The tobacco leaves were taken from the 9th to 10th leaf positions, and the entire root was selected for analysis. The sample information is shown in Table 1. All samples were cryodesiccated and ground in order to obtain a powder.

As the genomic analysis result, the mutant contained a frameshift mutation in the BBLs gene, and the mutation form contained either deletion of G or TAAG, as shown in Figure 1.

### 2.2. Alkaloids Analysis

The measurement of the tobacco alkaloids content was based on “YC/T 559-2018 Characteristic components of tobacco-Determination of alkaloids-Gas chromatography-mass spectrometry method and gas chromatography-tandem mass spectrometry method” [20]. A weight 0.3 g of the sample is placed into a 15 mL centrifuge tube, accurate up to 0.1 mg. Then, add 0.05 mL of methanol solution (quinoline 20 mg/mL, nornicotine 0.5 mg/mL, 2,3-bipyridine 0.5 mg/mL) as the internal standard solution and 2 mL of 5 wt% sodium hydroxide aqueous solution, respectively. After mixing and standing, add 10.0mL of extracting solution (dichloromethane: methanol (*v*/*v*) = 4:1), and extract by vortex shaking at a speed of 2000 r/min for 40 min; then, standing for 1hr, take the organic phase for testing.

Gas chromatography–mass spectrometry (ISQ 7000, Thermo Fisher, Waltham, MA, USA) was used to determine the tobacco alkaloids. The column used was a DB-35 GC column (30 m × 0.25 mm × 0.25 µm) (Agilent, Santa Clara, CA, USA). The inlet temperature was 250 °C and the helium gas flow rate through the column was 1 mL/min. The split ratio was 100:1 for nicotine and 10:1 for the other alkaloids. The initial oven temperature was held at 80 °C for 1 min, raised to 200 °C at a rate of 20 °C/min, then raised to 300 °C at a rate of 50 °C/min, and held for 8 min. The transfer line and the ion source temperatures were 280 and 250 °C, respectively. The ionization mode was the electron impact at 70 eV. The solvent delay was 5 min for nicotine and 7 min for the other alkaloids. The selected ion monitoring mode was used as ion scanning, and retention times, quantitation, and selected ion was accorded to the standard [20].

### 2.3. FT-IR and 2DCOS-IR Analysis

All the samples were measured in a Perkin-Elmer Spectrum One infrared spectrometer with a Deuterated Triglycine Sulfate (DTGS) detector(Perkin-Elmer, Waltham, MA, USA). The infrared spectra were collected from 4000 to 400 cm^−1^ with a resolution of 4 cm^−1^.

Each tobacco sample (about 1–2 mg) was blended with KBr (100 mg), grounded into powder, and then pressed into a tablet. Spectra were accumulated with 16 scans. The interferences of H_2_O andCO_2_ were subtracted when scanning. The test was repeated three times for each sample.

A temperature controller (CKW-1110, Beijing Chaoyang Automation Instrument Co., Beijing, China) was used to perform the thermal perturbation. The temperature range was from 50 to 120 °C with a heating rate at 2 °C/min, and IR spectra were collected at an interval of 10 °C.

IR spectra were processed by PE spectrum software (version 10.6.3). The 2DCOS-IR spectra were acquired using 2D correlation analysis software developed by Tsinghua University (Beijing, China).

The principal components analysis (PCA) was performed on MATLAB (version R2020b, Mathworks, Natick, MA, USA). The second derivative infrared (SD-IR) spectra were calculated by the 13-point Savitzky−Golay algorithm.

## 3. Results and Discussion

### 3.1. Effects of BBLs Knockout on Agronomic Traits

Tobacco is an important economic crop, and its agronomic traits will directly affect field cultivation, yield prediction, and the chemical composition of tobacco, so it has attracted the attention of growers and processors. Tobacco agronomic traits are an important means of tobacco breeding and an important evaluation method for the breeding value of gene-edited tobacco. At the flower-bud appearing stage, five agronomic characters, the plant height (PH), the girth of stem (GS), the number of leaves (NL), the length of top leaves (LTL), the width of top leaves (WTL), were analyzed according to “YC/T 142-2010 Investigation and measurement methods of agronomic characters of tobacco” (the results shown in Table 2) [21].

It can be found that, after the BBLs knockout, the PH, LTL, GS, WTL, and NL of tobacco decreased by 3.1%, 6.7%, 0.7%, 4.0%, and 2.6%, respectively, but the reduction was less than 10%. This indicates that the knockout of BBLs had an effect on tobacco growth, but the impact was limited. This may be because BBLs only regulate the downstream genes of nicotine and have little effect on growth. Since the knockout of BBLs had more negligible effect on the agronomic character, this mutant could be bred as a potential low-nicotine variety.

### 3.2. Effects of BBLs Knockout on Alkaloid Metabolism

Nicotine is composed of pyrrolidine and pyridine rings produced by independent metabolic pathways (Figure 2) [1,2,4]. The PIP family isoflavone reductase-like protein A622 and the berberine bridging enzyme-like protein BBLs are involved in the final stage of nicotine formation, namely the pyridine ring and pyrrole ring cross-linking reaction. The BBL enzyme is predicted to catalyze the final oxidation step of nicotine synthesis. The biosynthesis of nicotine, anatabine and anabasine, requires BBL. However, how the two rings are linked to form nicotine remains unclear.

As shown in Figure 3, in wild-type tobacco, the total alkaloids were higher in the leaves than in the roots, and these results are consistent with previous reports. In the mutants, this trend did not change, probably due to the knockout of BBL, which did not affect the alkaloids’ transport from root to leaf. After BBL knockout, the total alkaloids in the leaves and roots decreased by 60% and 74%, the nicotine decreased by 63% and 77%, 2,3-bipyridine decreased by 89% and 95%, and anabasine nicotine decreased 49% and 43%, anatabine decreased by 86% and 98%, and cotinine decreased by 3% and 89%, respectively, and these results are consistent with previous reports. Lewis and Kajikaw previously investigated the suppressing expression of BBLs tobacco in the field, greenhouse, and laboratory, respectively [2,4,9]. They found that the levels of anatabine and anabasine are concurrently reduced along with nicotine levels. These findings are consistent with our research. They speculated that this is because the BBL enzyme may be involved in the activation of nicotinic acid or the condensation reactions to produce bicyclic pyridine alkaloids, so the pyridine ring-containing nicotine, anatabine and anabasine, will be simultaneously reduced after the BBLs suppressing expression.

At the same time, nornicotine increased by 131% and 137%, and mysomine increased by 104% and 163%, respectively, and N-formylnornicotine increased by 3% and 338%, respectively. The results are exciting, as most of the alkaloid content was significantly reduced, but the content of nornicotine and nornicotine derivatives were increased steeply. This indicates that the synthesis of nornicotine may follow a new pathway process that is not catalyzed by the above six BBL enzymes, or that the knockout of the above six BBL enzymes may promote the conversion of nicotine to nornicotine and its derivatives. Lewis’ and Kajikaw’s studies also suggest that there may be an alternative nornicotine synthesis pathway that does not require BBLs [2,4,9]. Furthermore, further systematic research will be carried out in the future.

### 3.3. FT-IR Analysis of Typical Tobacco Samples

It is known that tobaccos are composed of polysaccharide substances, such as cellulose and lignin, calcium oxalate, alkaloids, proteins, and other more dilute compounds, all of which can be recorded in the IR spectra, whereas only components with higher abundance could be identified in the spectra and provide insightful information.

The infrared spectra of typical tobacco samples are shown in Figure 4, which shows that there are several important absorption bands: strong and broad absorption peaks at 3100–3800 cm^−1^, mainly caused by the hydroxyl (O-H) and amino (N-H) stretching vibrations of polysaccharides and proteins in the tobacco; at 2800–3050 cm^−1^ is the absorption of methyl and methylene stretching vibrations of the polysaccharides and proteins. The absorption band near 1628 cm^−1^ must be analyzed together with the peaks at 1316 and 779 cm^−1^, i.e., this peak contains calcium oxalate and the amide I of protein. The absorption peak near ~1400 cm^−1^ is due to the bending vibration of C-H in cellulose and lignin. The broad peaks at 800–1200 cm^−1^ are the characteristic absorption bands of polysaccharide substances, i.e., the C-O stretching vibration peaks of carbohydrates such as cellulose and lignin in tobacco (please refer to Table 3 for detailed spectral peak assignments). From the overall view of the spectrum, the roots and leaves of tobacco contain approximately the same composition, with the obvious difference being that the roots contain significantly more calcium oxalate than the leaf parts. Calcium oxalate is a secondary metabolite of the plant, and it is reasonable that the root content is higher than that of the leaves.

### 3.4. PCA Analysis of Roots and Leaves of Tobacco Samples

The purpose of building recognition models is to identify and classify the collected sample spectra and further apply them to the rapid identification of unknown samples. Since spectra are high-dimensional data, a combination of dimensionality reduction methods and pattern recognition algorithms is usually required, and commonly used methods include principal component analysis (PCA), partial least squares–discrimination analysis (PLS-DA), supporting vector machine (SVM), and artificial neural network (ANN) [25,26]. Among them, PCA is one of the most commonly used unsupervised methods.

The main process is to map the data from a high-dimensional space to a low-dimensional space using an orthogonal matrix to reduce the dimensionality of the data. The matrix form can be expressed as:(1)T=XW
where X denotes the original spectral data with a matrix size of I rows and J columns (I is the number of samples and J is the data dimension), W denotes the weight matrix of J rows and R columns, and T denotes the principal component matrix of I rows and R columns after dimensionality reduction.

The weight matrix is usually determined by using the maximization of variance principle to find the current optimal projection direction w in steps. The specific iterative process is as follows.

(a)Let the optimal projection direction be w. Then, the principal component vector is obtained after projection:
(2)t=Xw(b)The optimal projection direction should be such that the projection vector on it has the maximum differentiation, i.e., the optimization objective is to maximize the variance of t after projection; then, the problem is transformed to solve for the eigenvector w of the square matrix $X^{T}X$.
(3)$$\underset{||w||=1}{\max}\;\mathrm{var}(t^{T}t)=\mathrm{var}(w^{T}X^{T}Xw)$$(c)The projection X can be expressed in the form of a regression on the vector t:
(4)$$X=tp^{T}+E$$where p is the regression vector, E is the residual matrix, and t is the i-th principal component.(d)The residual matrix E is used as the new X. Repeat the process of (a)~(c) until the first R principal components are all determined.(e)After the first R principal components are selected, all vectors t form the principal component matrix T to complete the data dimensionality reduction.

Nontargeted PCA analysis was performed on the original spectra and second derivative spectra of the tobacco samples, and the distribution of the first two principal components PC1 and PC2 of the data were drawn on a two-dimensional plane graph, with the abscissa as PC1 and the ordinate as the ordinate for PC2. The spectral range of the principal component analysis is: 700–1800 cm^−1^. From the clustering results of the first two principal components, if the original spectrogram was used for direct PCA analysis, the infrared spectra of tobacco sample roots and leaves overlapped severely (Figure 5a). Using the second derivative spectra, the two can be completely distinguished (as shown in Figure 5b). However, as can be seen from Figure 5b, although the tobacco samples from the roots and leaves can be distinguished using the PCA method, Hongda and its BBLs knockout mutants overlapped severely and could not be distinguished.

### 3.5. 2DCOS-IR Spectroscopy and PCA Classification of Tobacco Samples

Dr. Isao Noda proposed the concept of perturbation-based two-dimensional correlation infrared spectroscopy first, and then extended this idea to other fields beyond infrared spectroscopy in 1993, establishing the concept of “Generalized Two-dimensional Correlation Spectroscopy” (2DCOS) [13,14]. The 2DCOS spectra are acquired by applying correlation analysis to a set of infrared spectra measured while the sample is perturbed by some physical or chemical stimulus.

The dynamic spectra are obtained when the original spectra of the perturbed sample are subtracted from a reference spectrum. Since the spectra are in practice recorded at discrete steps over the entire perturbation, the 2DCOS spectra may be calculated according to some simple procedures. When the spectra of the sample are measured at m steps with an equal interval of perturbation t varying from T_min_ to T_max_, the dynamic spectral intensities at variables ν may be expressed as a column vector. The synchronous correlation intensity may be calculated as:(5)$$\Phi \left(\nu_{1},\nu_{2}\right)=\frac{1}{m-1}y{\left(\nu_{1}\right)}^{T}\cdot y\left(\nu_{2}\right)$$

Then, the two-dimensional correlation synchronous spectrum is obtained by plotting the correlation intensity value Φ (ν_1_, ν_2_) against variables ν_1_ and ν_2_. Since there are two independent variables for each correlation intensity value, the 2DCOS spectra may be shown as contour maps. Peaks in the synchronous 2DCOS spectrum represent the coincidence of the spectral intensity variations at corresponding variables along the perturbation. The spectral intensity variation at a variable is always the same as itself, so there are only positive peaks, which are defined as the autopeaks, along the diagonal of the synchronous 2D correlation spectrum. The perturbation applied to the sample changes the intra- or intermolecular interactions of the sample. By analyzing the spectral changes, information about the intramolecular functional group interactions and intermolecular interactions can be obtained.

Based on the above advantages of the 2D correlation spectroscopy in analyzing the complex mixture systems with an enhanced signal resolution, we performed 2D correlation spectroscopy on the above 28 samples of tobacco using temperature as a perturbation, expecting to distinguish the samples before and after gene editing with the higher apparent resolution of 2D correlation spectroscopy.

In the synchronous 2DCOS-IR spectra of the tobacco root samples (Figure 6), Hongda and its BBLs knockout mutants have three distinct auto peaks, located at around 1450, 1580, 1640, and 1730 cm^−1^, respectively. As seen in Figure 4, the absorption peak of calcium oxalate at 1620 cm^−1^ is a broad peak, and the rising and falling segments of this peak are located near 1580 and 1640 cm^−1^, respectively. The perturbation we used in the two-dimensional correlation spectroscopy analysis is temperature, and the autopeaks from the synchronous 2DCOS spectra show that these two bands respond more strongly to thermal, thus forming two autopeaks. The autopeak at 1450 cm^−1^ is attributed to the C-H deformation in lignin and carbohydrates, and the autopeak at 1730 cm^−1^ is attributed to carbonyl and aliphatic groups [16]. It can be seen that the 2DCOS-IR spectra of Hongda and its BBLs knockout mutants are very similar.

Since it is difficult to perform PCA clustering analysis for the synchronous 2D spectra of Hongda and its BBLs knockout mutants directly, we chose the autopeak curves as the compromise solution. In Noda’s rule, the autopeaks are the sensitivity of functional groups or spectral bands to the perturbation, which can reflect the change of sample composition under perturbation, so it is feasible to use autopeaks for PCA clustering analysis. Figure 7 shows the results after clustering the root samples of Hongda and its BBLs knockout mutants, and from the figure, it can be seen that only one sample was misclassified, and the classification rate was >90% correct. Similarly, we also performed PCA analysis on the leaf part of Hongda and its BBLs knockout mutants. Additionally, it can be seen that there is only one sample misclassified. The present results demonstrate the effectiveness of the autopeaks of the 2DCOS-IR spectra for the classification of Hongda and its BBLs knockout mutants’ samples.

## 4. Conclusions

In this paper, the agricultural traits, alkaloids content, FT-IR, and 2DCOS-IR spectroscopy analysis of the tobacco after BBLs knockout were studied. The BBLs knockout has little effect on tobacco agronomic traits. In the mutants, most alkaloids, such as nicotine, are significantly reduced, but the content of myosmine and its derivatives increases dramatically. Since BBLs is only the downstream gene of nicotine biosynthesis, it has limited effect on tobacco metabolism. Hence, the FT-IR can only distinguish the tobacco leaf from the root, but not identify mutants from wild-type tobacco. Instead, the PCA analysis of the autopeaks of 2DCOS-IR clearly distinguishes the mutants from the wild-type tobacco, with an accuracy rate of more than 90%. The high resolution 2DCOS-IR has great potential for application in the nondestructive screening and identification of gene-editing materials.

## Figures and Tables

**Figure 1 molecules-27-03817-f001:**
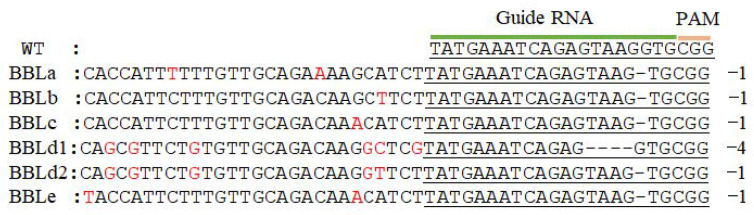
CRISPR/Cas9-mediated BBLs gene mutation in *Nitotiana tobacum*.

**Figure 2 molecules-27-03817-f002:**
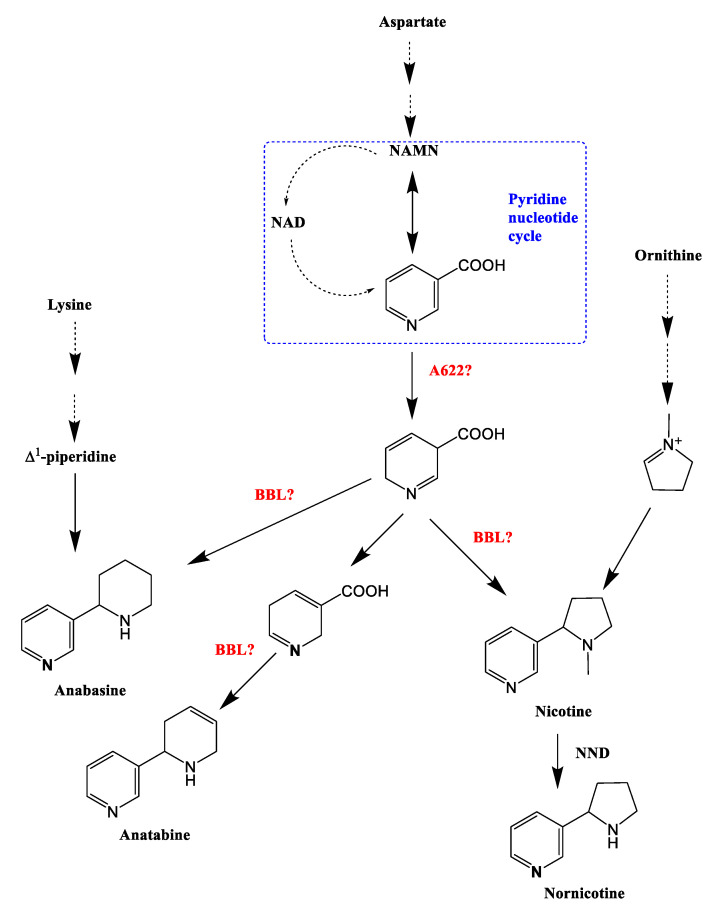
Schematic diagram of alkaloid biosynthesis in *Nitotiana tobacum*.

**Figure 3 molecules-27-03817-f003:**
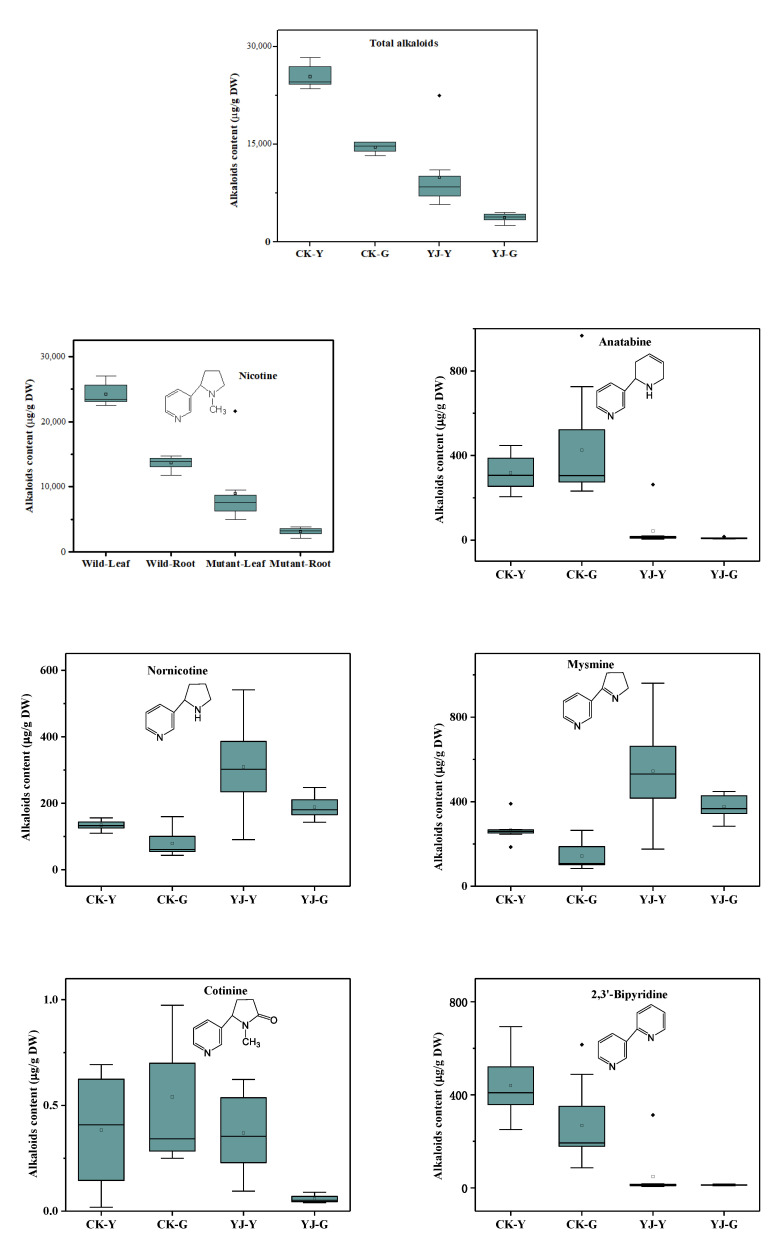
Comparison of alkaloids content of Hongda and its BBLs knockout mutants.

**Figure 4 molecules-27-03817-f004:**
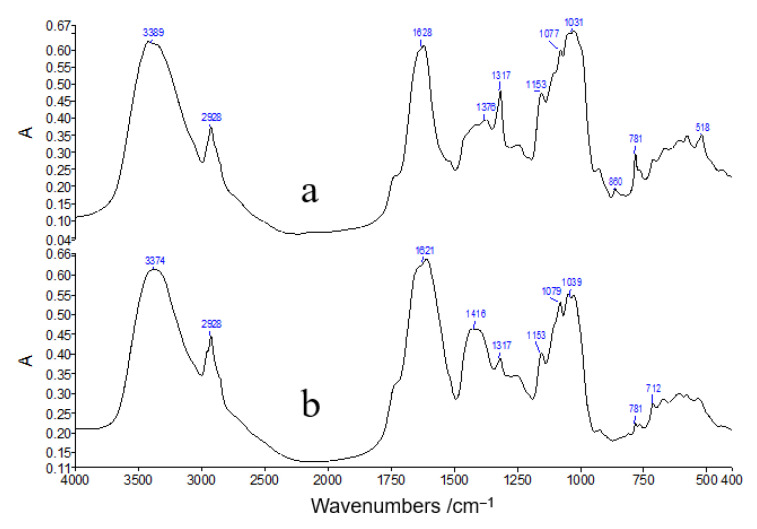
Typical FT-IR spectra of tobacco samples. (**a**) Root; (**b**) Leaf.

**Figure 5 molecules-27-03817-f005:**
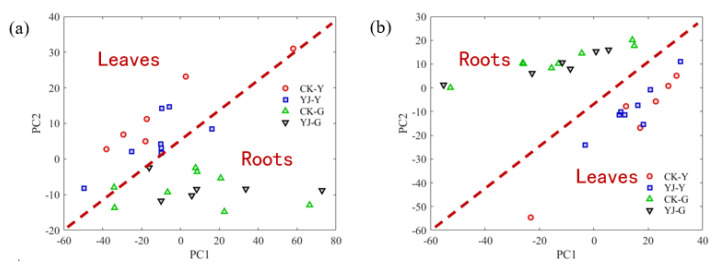
PCA results performed on the original spectra (**a**) and second derivative spectra (**b**) of tobacco samples.

**Figure 6 molecules-27-03817-f006:**
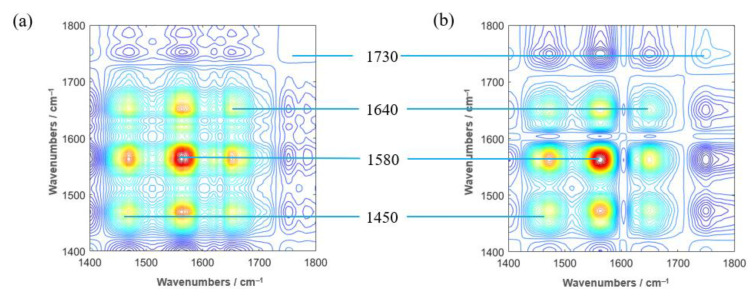
Synchronous 2DCOS−IR spectra of Hongda (**a**) and its t BBLs knockout mutants (**b**) in the range of 1400–1800 cm^−1^.

**Figure 7 molecules-27-03817-f007:**
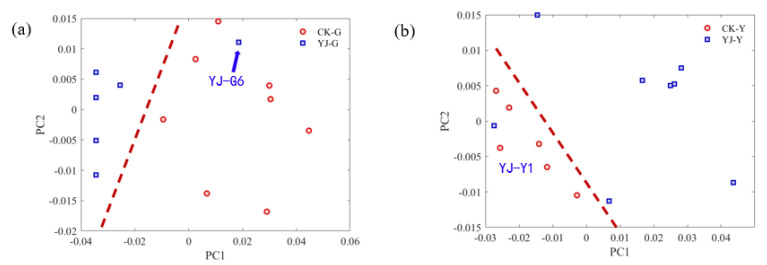
PCA results performed on the autopeaks of synchronous 2DCOS spectra of Hongda and its BBLs knockout mutants’ samples. (**a**) Roots and (**b**) leaves.

**Table 1 molecules-27-03817-t001:** The data of the sample information.

	Analysis Part	Code Name	Sample Numbers
Hongda wild-type tobacco (CK)	Root	CK-G	8
	Leaf	CK-Y	6
Gene knockout mutants (YJ)	Root	YJ-G	6
	Leaf	YJ-Y	8

**Table 2 molecules-27-03817-t002:** The agronomic traits data of Hongda and its BBLs knockout mutants.

ID		PH (cm)	GS (cm)	LTL (cm)	WTL (cm)	NL
	Max	130.0	14.0	68.0	25.0	15.0
YJ	Ave	126.8	13.3	61.4	24.0	14.8
	Min	110.0	12.8	56.0	19.0	14.0
	Max	134.0	13.8	67.0	28.0	16.0
CK	Ave	130.8	13.4	65.8	25.0	15.2
	Min	128.0	11.8	62.0	24.0	15.0

**Table 3 molecules-27-03817-t003:** Tentative assignments of FT-IR of tobacco samples [22,23,24].

Peak Position/cm^−1^		Functional Group	Main Attribution
Root	Leaf		
3389	3374	sν (O−H, N−H)	OH, NH
2928	2928	ν_as_ (CH_2_)	CH_2_
1628	1621	ν (C=O−O), ν (Ar)	calcium oxalate, carboxyl
-	1416	δ (CH_2_), δ (C−O−H)	lignin, cellulose
1375	-	ν_as_ (C−N−C), δ (C−O−H)	lignin, cellulose
1317	1317	ν (C−O−H)	calcium oxalate
1153	1153	ν (C−C,C−O), δ (C−O−H)	cellulose
1077	1079	ν (C−C,C−O), δ (C−O−H)	lignin
1031	1039	ν (C−C,C−O), δ (C−O−H)	cellulose
850		ν (C−C,C−O), δ (C−O−H)	cellulose
781	781	ν (C−C)	calcium oxalate
518	-	C−O−H	calcium oxalate

Notes: ν: stretching. ν_s_: symmetrical stretching. ν_as_: asymmetrical stretching. δ: bending.

## Data Availability

Not applicable.

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
