# Peer review of "The Agronomic Traits, Alkaloids Analysis, FT-IR and 2DCOS-IR Spectroscopy Identification of the Low-Nicotine-Content Nontransgenic Tobacco Edited by CRISPR–Cas9"

_molecules, 2022, doi:10.3390/molecules27123817_

Round 1
Reviewer 1 Report
The paper studies the differences in some agronomic traits of wild and mutant tobacco. The morphological traits of wild and mutant tobacco plants are described, and the difference in the production of some alkaloids is discussed. Additionally, a principal component analysis is presented using the IR spectra of the wild and BBLs knockout tobacco samples. It is noteworthy that it was possible to identify that, in the treated samples, the production of other alkaloids increases while the nicotine concentration decreases. On the other hand, PCA of the 2DCOS spectra made it possible to determine whether the tobacco is wild or genetically modified.
The article has many valuable aspects to highlight that deserves to be published. However, some important details need to be corrected before publication.
- I think the title can be improved to highlight the findings obtained from the study carried out.
- Regarding the analysis of the agronomic traits of the plants, the description of the results is very general and does not agree with the results presented in Table 2 (lines 133-135, Table 2). While the description of the results indicates that after treatment with enzymes, the values ​​of the five studied traits decrease, the opposite is observed in four of the five traits in the table. That is, the description is not consistent with the results presented. It is recommended to include a brief description of the difference in each agronomic trait's behavior.
- In line 150, it is stated that “all alkaloids were higher in leaves than in roots”. This is not true in all cases. The authors may be referring to total alkaloids, but this is not true for all alkaloids. In some cases, it is not possible to say that there is a difference in the concentration of alkaloids for wild versus mutant tobacco since the error bars of their graphs indicate that there is no difference; such is the case of natabine, nornitoctine and anabasine. It is necessary to correct this aspect in the description of the article.
- On the other hand, it is mentioned that the results of this article are consistent with the results reported in other papers (lines 150-164); however, no references or comparison data are mentioned. It is necessary to support the discussion of the results with arguments from other investigations.
- Finally, it seems that, as a great contribution to the paper, it is pertinent that a step-by-step description of the PCA process is given. This can be a very helpful tool for young researchers who would like to apply this tool.
Reviewer 2 Report
Please see attached file.

Reviewer 3 Report
The manuscript molecules-1735792, entitled “The Agronomic Traits, Alkaloids Analysis, FT-IR and 2DCOS-IR Spectroscopy Identification of the Low Nicotine Content Nontransgenic Tobacco Edited by CRISPR-Cas9” is a good paper, with interesting applications. I suggest to accept it, after trivial typo corrections listed below.
Line 56: reference 10 is in superscript, while it should be written [10]
Line : ERRATA: “” CORRIGE: “”
Line 57: ERRATA: “can significantly improves” CORRIGE: “can significantly improve”
Line 74: ERRATA: “the sampling use” CORRIGE: “the sampling used”
Line 92: ERRATA: “Extract the solution” CORRIGE: “of extracting solution”
Line 108: please specify the acronym DTGS
Line 112: insert space after the word “and”
Line 120: ERRATA: “principle component analysis” CORRIGE: “principal components analysis”
Line 136: ERRATA: “has” CORRIGE: “have”
Line 158: ERRATA: “104%, respectively %, 163%” CORRIGE: “104% and 163%, respectively”
Line 161: ERRATA: “It indicates that the synthesis of nornicotine may exist a new pathway process that does not catalysis by the above six BBL enzymes” CORRIGE: “It indicates that the synthesis of nornicotine may follow a new pathway process that is not catalysed by the above six BBL enzymes”
Figure 4: move the labels “a” and “b” in order to prevent their superimposing to the spectrum
Line 282: ERRATA: “And, ” CORRIGE: “Instead, ”
Round 2
Reviewer 1 Report
After reviewing the corrected versión of the manuscript, I consider it cover the scientific quality to be published in Molecules.
Only i have a suggestion: since in the update versión of the manuscript it is included some equations, i believe is necesary numering the equation in line 271 (page 10)
Author Response
Comments
Only i have a suggestion: since in the update versión of the manuscript it is included some equations, i believe is necesary numering the equation in line 271 (page 10)
As suggested, the number of the equation is added, please see line 271.